# Effect of surface water and underground water drip irrigation on cotton growth and yield under two different irrigation schemes

**Nihal Niaz**[☯], **Cheng Tang**[iD][☯]*

College of Agronomy, Shihezi University, Shihezi City, Xinjiang Province, China

☯ These authors contributed equally to this work.
* tangcheng1983@163.com

**Data Availability Statement:** All relevant data are within the article and its Supporting information files.

**Funding:** This work was supported by the Key Technologies Research and Development Program

## Abstract

To investigate the effect of surface water and underground water drip irrigation on cotton yield, dry matter accumulation and nutrients uptake, two consecutive field experiments were conducted. The first experiment (different mixing ratio irrigation) comprised of five ratios of underground water to surface water including; 1:0 (U), 0:1 (S), 1:1 (U:S = 1:1), 1:2 (U:S = 1:2) and 1:3 (U:S = 1:3). Whereas, the second experiment (round irrigation) comprised of eight treatments including: 1:3 ($T_1$), 2:2 ($T_2$), 3:1 ($T_3$), {S:U 3:1 ($T_4$)}, 2:2 {S:U ($T_5$)}, 1:3 {S:U ($T_6$)}, 4:0 ($T_7$) and 0:4 ($T_8$). The average concentration of leaves dry matter after 8th irrigation in different mixing ratio experiment was significantly increased by 131.2% (S), 34.4% (U: S = 1:1), 59.3% (U: S = 1:2), and 93.7% (U: S = 1:3), respectively, relative to U treatment. Likewise, the stem dry matter increased from 48.5 g (U), to 122.2 g (S) and 101.6 g (U:S = 1:3). The soil available N at 0–20 cm after 8th irrigation recorded an average increase rate of 40.1%, 6.6%, 13.5%, and 29.5%, respectively. However, at 20-40cm an average increase rate of 37.4% (S), 7.1% (U: S = 1:1), 20.0% (U: S = 1:2), and 21.9% (U: S = 1:3) were noted ($p < 0.05$). The highest cotton yield of 6571 kg h$^{-1}$ was recorded in S treatment compared with the U treatment (5492 kg h$^{-1}$), U: S = 1:1 (5502 kg h$^{-1}$), U: S = 1:2 (5873 kg h$^{-1}$) and U: S = 1:3 (6111 kg h$^{-1}$). Contrastingly, in round irrigation experiment the highest leaves dry matter at various growth stages were recorded in T8 treatment. For instance, compared with T7 treatment an average increase rate of 50.6% (growth), 100.9% (boll) and 93.3% (boll opening), in stem dry matter were recorded in T8 treatment. Moreover, the concentration of N in round irrigation at 0–20 cm at different growth stages were 83.3±2.8 (growth stage), 79.01 ±1.84 (boll stage), and 96.16±3.83 (boll opening stage) in T8. Whereas, in T7 the concentration of N was 36.1±5.9 (growth), 54.51±2.81 (boll), and 53.9±3.83 (boll opening) ($p < 0.05$). Similarly, cotton yield were substantially higher in T8 applied treatment and follows the sequence of T8 > T1 > T4 > T2 > T5 > T3 > T6 > T7. Overall, our findings provide meaningful information to current irrigation practices in water scarce regions. Improving water use efficiency is a viable solution to the water scarcity. Therefore, surface water irrigation is recommended as an effective irrigation strategies to improve cotton yield and growth.

of China (2017YFC0504302-02). The funders had no role in study design, data collection and analysis, decision to publish, or preparation of the manuscript.

**Competing interests:** The authors have declared that no competing interests exist.

## 1. Introduction

Water for irrigation is one of the most limiting factor for future global agricultural developments [1, 2]. Climate change, and over-exploitation of water resources in arid and semi-arid regions of the world are being subjected to severe water shortages [3]. Consequently, water scarcity threatens the sustainability of irrigated cotton production in many regions around the world. To overcome the inherent problem, efficient use of water to cotton crop is an important consideration where irrigation water resources and rainfall are limited [4]. Moreover, effective use of irrigation is not simply a water saving irrigation but it is a comprehensive exercise to lower down the competition for fresh water among municipal, industrial and agricultural sectors in several countries in the world especially in China [5]. Therefore, optimizing water use for cotton production by introducing new irrigation strategies as well as efficient management of an irrigation network to optimize the problem of water shortage and ecological environment deterioration is of paramount importance [6].

Cotton (*Gossypium hirsutum* L.), an important source of natural fibers for textile industries that serve the humanity from at least more than four to seven thousand years ago [7]. Its current global production is estimated to be 24.65 million tons, with 6.71 million tons produced in America, 0.38 million tons produced in Europe, and 15.06 million tons produced in Asia [8]. Following this, only China accounts for one-quarter of the world's cotton output and one-third of the world's cotton consumption [9]. On the flip side, cotton production is completely dependent on sufficient irrigation i.e., the shortage of irrigation water resources restrict the comprehensive improvement of cotton productivity [10, 11]. It is estimated that approximately 700 to 1200 mm water are required for cotton growth during their growing season, depending on irrigation method, and production goals [12]. Several irrigation management strategies have been discussed to enhance water-use efficiency of cotton crop in recent years. For instance, a study conducted by Grismer [13] noted that, in Arizona counties, for upland cotton actual evapotranspiration (ETc) water-use efficiency varied from 1.27 to 1.38 kg/ha-mm while, for pima cotton, it varied from 0.9 to 1.09 kg/ha-mm. In California counties, ETc water-use efficiency varied from 1.34 to 2.10 kg/ha-mm and 1.51–1.77 kg/ha-mm for upland and pima varieties, respectively.

Surface water resources (water from rivers and reservoirs) and groundwater resources (water stored in aquifers) were used for agriculture purposes in arid and semi-arid regions. However, irrigation with underground water has a negative impact on cotton productivity, plant nutritional condition, and dry matter accumulation. For instance, excessive underground water irrigation exacerbates the soil salinization problems, and reduce crop yield [11, 14]. In parallel, well water irrigation with low temperature potentially inhibits the growth and development of cotton plant [15]. Subsequently, low soil temperatures may slow down the uptake rate of nutrients such as N so much that they turn out to limit vegetative growth rates [16]. Following this at 20°C-RZT, nutrient concentrations have been significantly affected plant growth indexes, indicating that low root temperature inhibited high nutrient effects on plant growth [16, 17]. On the other hand, surface water irrigation shows some promising effect on increasing crops yield [18, 19]. For example, irrigating seedlings with warm water (surface water) can increase the stem thickness, leaf area, root coefficient, photosynthetic rate, dry matter per unit fresh weight, and root-shoot ratio [16]. Therefore, it is critical to understand how different irrigation regime effect cotton growth and productions under limited water supplies.

Xinjiang Uygur Autonomous Region is one of the most water-scarce states in China. According to the Statistics Bureau of Xinjiang Uygur Autonomous Region, the water production per unit area of Xinjiang Uygur Autonomous Region is 51 mm, which is the second highest rank in the country [19]. The large number of glaciers in Xinjiang Uygur Autonomous

Region and the small unit area are important issues of water resources in Xinjiang Uygur Autonomous Region. Snowmelt water from glaciers accounts for more than 25% of total surface water [20]. In the present-day context, lot of emphasis is being given on improvements in irrigation practices to increase crop production and to sustain the productivity levels. In this study, two field experiments were carried out in calcareous soils. We hypothesize that surface water drip irrigation outcompetes underground water irrigation in increasing cotton yield, dry matter accumulation and NPK uptake. Therefore, the objectives of this study were to (i) compare the effects of different irrigation schemes on cotton growth and yield. (ii) Clarify the response of cotton growth to surface and underground water application, (iii) finally put forward an appropriate irrigation strategy to increase cotton yield in calcareous soil.

## 2. Materials and methods

### 2.1 Experimental site

The study area is located in the Mosuowan reclamation region (44°03′N, 86°05′E), which is located on the northern slope of Tianshan Mountain in Xinjiang and is surrounded by the Gurbantunggut Desert. According to WRB soil taxonomy, the tested soil is classified as Calcisol Fluvisols. The study area has a typical continental climate with a mean annual precipitation of 115 mm, and rainfall largely occurs from April to July. The mean annual potential evapotranspiration is approximately 2,000 mm. The physicochemical characteristics of given soil is listed in (Table 1).

### 2.2 Experimental design

**2.2.1 Experiment I: Different mixing ratio experiment 2019.** A three replicated Randomized Complete Block Design (RCBD) was employed to layout the experiment having plot size of 55.2 m$^2$. The first experiment was a field experiment conducted on April 12, 2019 with five ratios of underground water to surface water including; 1:0 (U), 0:1 (S), U:S = 1:1, U:S = 1:2, and U:S = 1:3, respectively. To mix underground water and surface water at a specific ratio, we dig several wholes and cover it with plastic, after that we supplied surface and underground water into the whole with a constant water ratio by using water ratio measurement meter. After getting our specific ratio of underground and surface water we mixed the

**Table 1. Selected physical and chemical properties of the tested soils.**

| Soil properties | 0-20cm |
|---|---|
| pH | 8.83±0.03 |
| OM (g kg$^{-1}$) | 13.7±0.23 |
| Total-N (g kg$^{-1}$) | 1.20±0.05 |
| Available-P (mg kg$^{-1}$) | 17.5±3.52 |
| Total-P (g kg$^{-1}$) | 1.08±0.12 |
| Total-K (mg kg$^{-1}$) | 21.0±2.07 |
| Available-K (mg kg$^{-1}$) | 241.4±0.00 |

Data were presented as the mean ± standard error (SE), n = 3 at a significance level of $p < 0.05$.

pH was determined at soil to milli-Q water ratio of 1:5 w/v using pH meter.

Organic matter was measured by potassium dichromate volumetric method (Shaw, 1959) [21].

Total N was measured by the semimicro-Kjeldahl method [22].

Total P was measured by the perchloric acid digestion method [23].

Total and available K was measured by the flame photometry method [22].

**Table 2. The amount of nutrients (fertilizer) and water during the period of cotton irrigation in different mixing ratio 2019.**

| Irrigation time | Irrigation water ($m^3.ha^{-1}$) | Nitrogen ($Kg.ha^{-1}$) | Phosphorus ($Kg.ha^{-1}$) | Potassium ($Kg.ha^{-1}$) |
|---|---|---|---|---|
| 12.6.2019 | 450 | 30 | 15 | 15 |
| 20.6.2019 | 450 | 75 | 75 | 75 |
| 30.6.2019 | 375 | 105 | 105 | 105 |
| 10.7.2019 | 375 | 105 | 105 | 105 |
| 20.7.2019 | 300 | 75 | 150 | 150 |
| 31.7.2019 | 300 | 30 | 180 | 180 |
| 10.8.2019 | 225 | 75 | 75 | 75 |
| 24.8.2019 | 225 | 45 | 45 | 45 |

nutrients (fertilizers) in the water and supplied it to the cotton field through by drip irrigation. The schematic of our experimental design are presented in S1 Fig. The details supply of water and nutrients for each irrigation in experiment first has listed in (Table 2).

**2.2.2 Experiment II: Round irrigation experiment 2020.** The experiment II (round irrigation) was a field randomized complete block design (RCBD) having plot size of 55.2 $m^2$. Treatments includes, 1:3 ($T_1$), 2:2 ($T_2$), 3:1 ($T_3$), {S:U 3:1 ($T_4$)}, 2:2 {S:U ($T_5$)}, 1:3 {S:U ($T_6$)}, 4:0 ($T_7$) and 0:4 ($T_8$). The treatment ratios represents the supply of surface and underground water in different stages of cotton growth i.e., seedling stage, growth stage, boll stage and boll opening stage. In round water irrigation we supplied the constant ratios of both surface water and underground water directly to the cotton field at various growth stages prior before mixing it. For maintaining the required ratios we first supplied the specific ratio of surface water and then we supplied the specific ratio of underground water by using a water ratio measurement meter. Further details about round irrigation is given in (Table 3). Nutrients (fertilizer) and water were supplied through drip irrigation. The supply of water and nutrients for each irrigation has listed in (Table 4).

## 2.3 Soil and plants sampling

Soil samples were collected at depths of 0–20 and 20–40 cm from each block after 3–5 days of irrigation. Samples were air dried, sieved through 1mm and 0.15 mm for nutrients determination. The pH was determined at soil to milli-Q water ratio of 1:5 w/v using pH meter. The organic matter was determined by Walkley-Black chromic acid wet oxidation method [21]. Nitrogen was determined by semimicro-Kjeldahl method [22]. Phosphorus was measured by

**Table 3. Supply of surface and underground water at different growth stage of cotton during round irrigation 2020.**

| Treatments | Seedling stage | Growth stage | Boll stage | Boll opening stage |
|---|---|---|---|---|
| T1 U:S (1:3) | [1]U | S | S | S |
| T2 U:S (2:2) | U | U | S | S |
| T3 U:S (3:1) | U | U | U | S |
| T4 S:U (3:1) | [2]S | S | S | U |
| T5 S:U (2:2) | S | S | U | U |
| T6 S:U (1:3) | S | U | U | U |
| T7 U:S (4:0) | U | U | U | U |
| T8 U:S (0:4) | S | S | S | S |

[1] U represents the supply of underground water.

[2] S represents the supply of surface water.

**Table 4. The supply of water and nutrients (fertilizer) for each irrigation (round irrigation 2020).**

| Irrigation order | Water (m$^3$.ha$^{-1}$) | Nitrogen (Kg.ha$^{-1}$) | Phosphorus (Kg.ha$^{-1}$) | Potassium (Kg.ha$^{-1}$) |
|---|---|---|---|---|
| 1 | 450 | 30 | 15 | 15 |
| 2 | 450 | 75 | 75 | 75 |
| 3 | 375 | 105 | 105 | 105 |
| 4 | 375 | 105 | 105 | 105 |
| 5 | 300 | 75 | 150 | 150 |
| 6 | 300 | 30 | 180 | 180 |
| 7 | 225 | 75 | - | - |
| 8 | 225 | 45 | - | - |

the Perchloric acids digestion method [23]. Potassium was determined by flame photometry [22]. Available NPK and organic matter were determined in soil samples after growth stage, boll stage and boll opening stage.

Plant samples were randomly collected from each block at interval of 3–5 days after each irrigation. Plant samples were divided into the following parts (leaves, stems, roots and fruits), washed with tap water and then dried in oven at 105°C for 30 minutes and then at 75°C for 3 days. The plant samples were then weighed with balance and the dry matter data were calculated after growth stage, boll stage and boll opening stage (Fig 1), following the method of [24].

### 2.4 Statistical analysis

Data were analyzed using the SPSS 25.5 statistical program (SPSS Inc., Chicago, IL, USA) with ANOVA for various growth stages as dependent on surface water ($S$), underground water ($U$) and their interaction ($S^*U$) at a significance level of $p < 0.05$. Moreover, a Duncan multiple range test was carried out to test the significant differences between different treatments. GraphPad Prism 12.0 software (GraphPad Software, Inc., San Diego, CA, USA) was used for data processing and images making. All results in figures and tables were presented as mean ± standard error (SE) of three replicates, and a significance level of $P < 0.05$ was used for all analysis.

## 3. Results

### 3.1 Effects of different mixing ratio irrigation on the dry matter accumulation

The average dry matter accumulation in the different parts of the cotton plant fluctuated greatly and increased sharply after each irrigation time (Fig 2). For instance, the significant maximum dry weights (DW) were recorded in S treatment followed by the U: S = 1:3 treatment. However, the lowest dry matter accumulation was recorded in the U treatment. The concentration of leaves dry matter after 8$^{th}$ irrigation increased from 32.8 ± 0.15 g (U), to 74.1 ± 0.27g (S), 43.5 ± 0.18g (U:S = 1:1), 51.1 ± 0.21g (U:S = 1:2) and 62.9 ± 0.37g (U:S = 1:3), with an average increase rate of 131.2%, 34.4%, 59.3%, and 93.7%, respectively (*P < 0.05*, Fig 2a). Compared with underground water irrigation a tremendous average increase rate of 131.2% and 93.7% were noted with the application of surface water and different mixing ratios. Similarly, stem dry matter of five applied treatments as affected by eight irrigation regimes are presented in Fig 2b. The concentration of stem dry matter after 8$^{th}$ irrigation increased from 48.5 g (U), to 122.2 g (S) and 101.6 g (U:S = 1:3), with an average increase rate of 151.5% and 109.5%, (Fig 2b). Moreover, compared with underground water irrigation treatment (24.6 g),

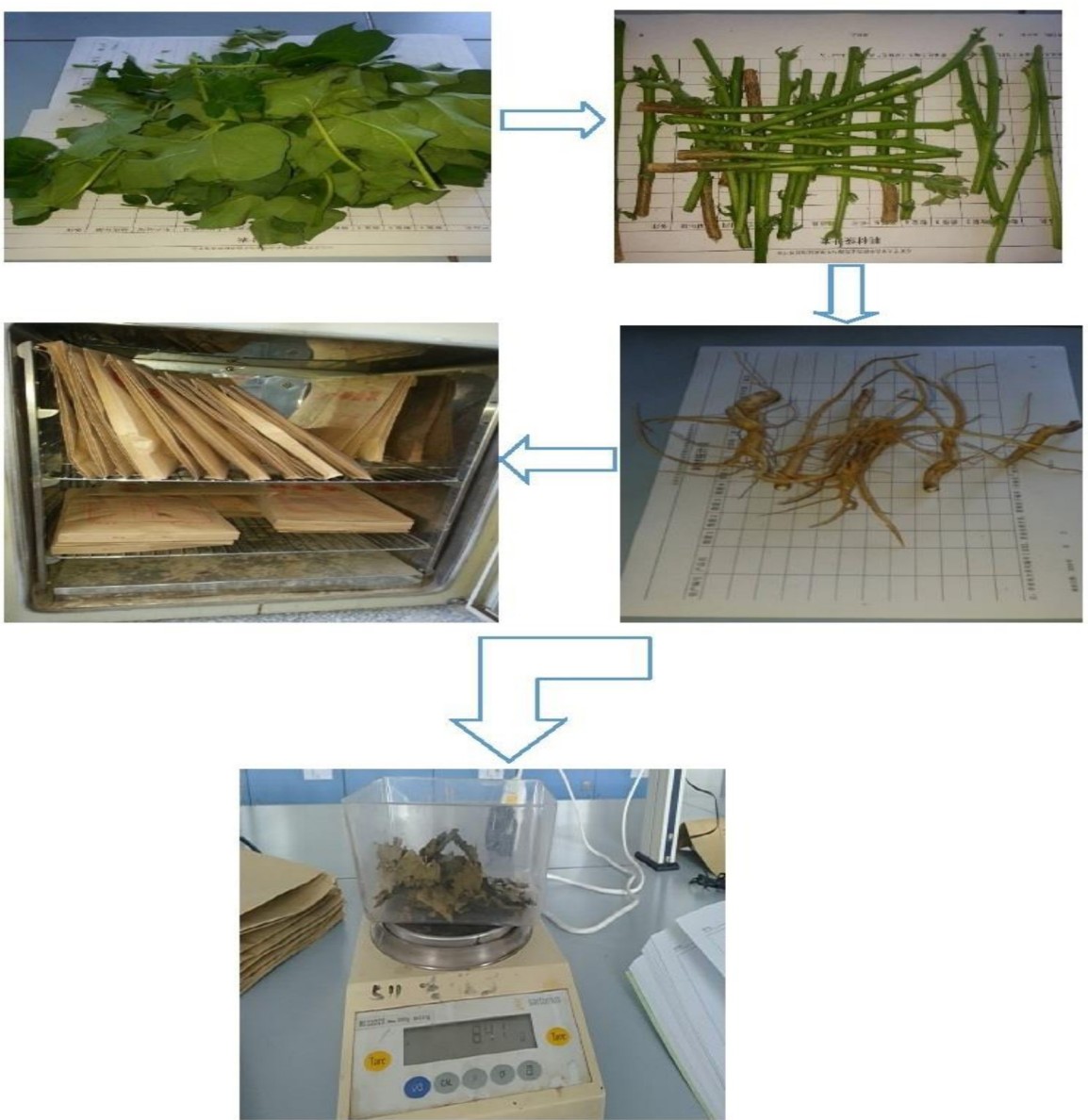

**Fig 1. Schematic of dry matter accumulation calculation.**

a significant highest root dry matter was recorded with surface water treatment (51.74 g) and U: S = 1:3 treatment (46.96 g) ($p < 0.05$, Fig 2c). Similarly, compared with surface water, underground water significantly suppressed roots dry matter accumulation and follows the sequence of S> U: S = 1:3> U: S = 1:2> U: S = 1:1>U. The highest fruits dry matter accumulation was recorded in the treatment of surface water (83.43 ± 0.34 g) relative to all other applied treatments ($p < 0.05$, Fig 2d). Furthermore, correlation and regression analysis was performed to explore the relationship between soil available nutrients and dry matter accumulation (Fig 3). It can be seen from the figure that there is a strong positive correlation between soil available nutrients and dry matter accumulation in different mixing ratio irrigation (2019).

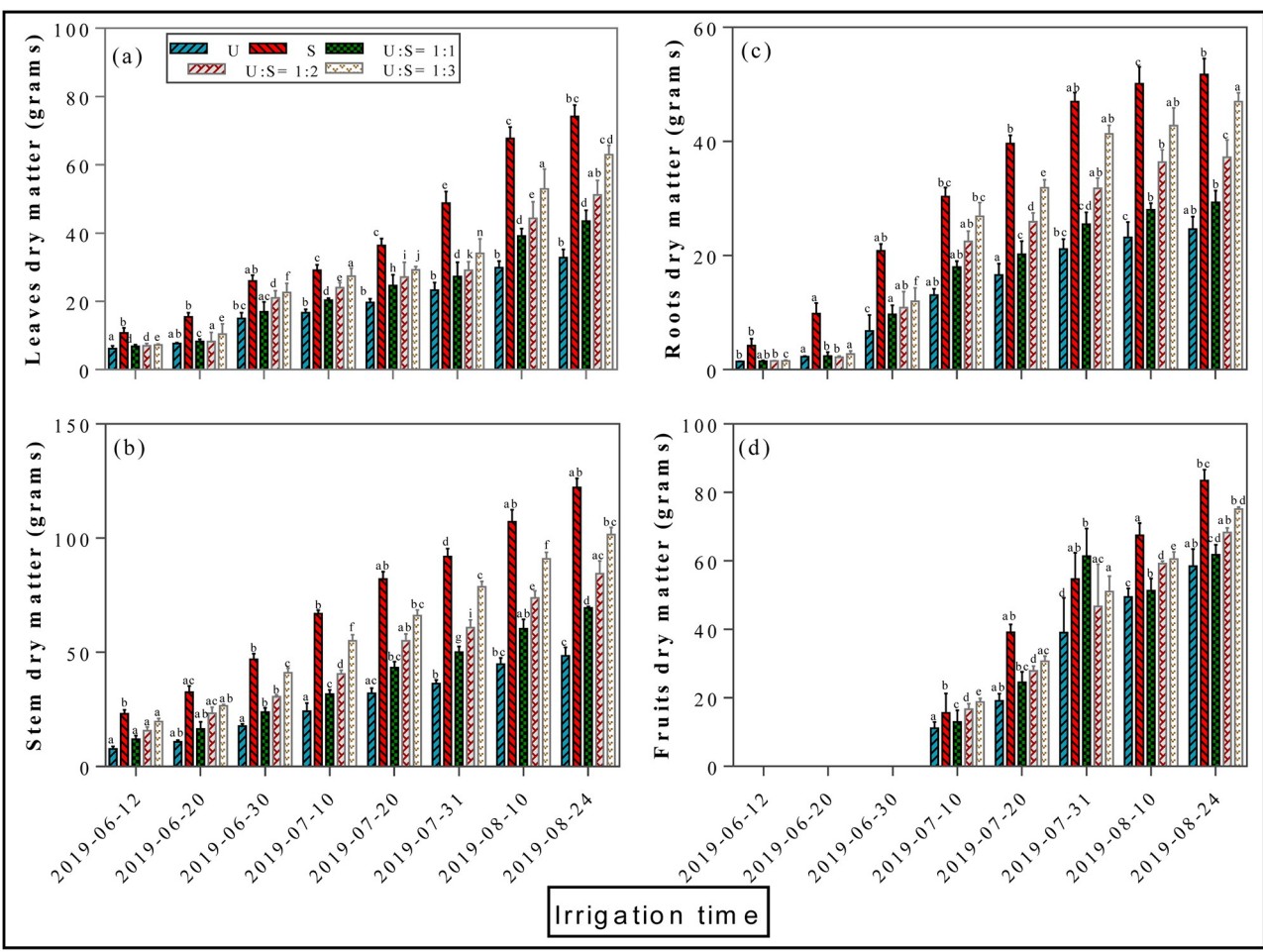

**Fig 2. Effects of different mixing ratio irrigation methods on the accumulation dry matter: Fig (2a) represents leaves dry matter; (2b) represents stem dry matter; (2c) represent roots dry matter; and (2d) represent fruits dry matter, respectively.** Data were presented as the mean ± standard error (SE) of three replicates at a significance level of p < 0.05 (based on ANOVA).

## 3.2 Effects of round irrigation on dry matter accumulation

Round irrigation also significantly affected dry matter accumulation in the different parts of the plant at various stages of growth ($p < 0.05$, Fig 4). The biomass of yield-related organs showed a trend of gradual increase with the progression of the growth period, and the most intense change was at the beginning of the boll development stage. Maximum dry weights (DW) were achieved at boll opening stage and minimum at growth stage. The DW of leaves, stem, roots and fruits continued to increase initially and then declined and stayed stable comparatively because of leaf senescence and termination of reproductive development at final stages (Fig 4a–4d). The highest leaves dry matter of 14.4g (growth stage), 88.8g (boll stage) and 100.7g (boll opening stage) were observed in T8 treatment compared with the lowest, 8.9g (growth stage), 42.9g (boll stage) and 78.7g (boll opening stage) of T7 treatment. Moreover, results of two-way ANOVAs reveals a significant main and interactive effect on various stage of leaves dry matter treated with surface water ($p < 0.05$, Fig 4a). Compared with T7 applied treatment an average increase rate of 50.6%, 100.9% and 93.3%, at growth, boll and boll opening stage respectively, in stem dry matter were recorded in T8 treatment. Likewise, a significant main and interactive effect were observed at various stages of stem dry matter ($p < 0.05$,

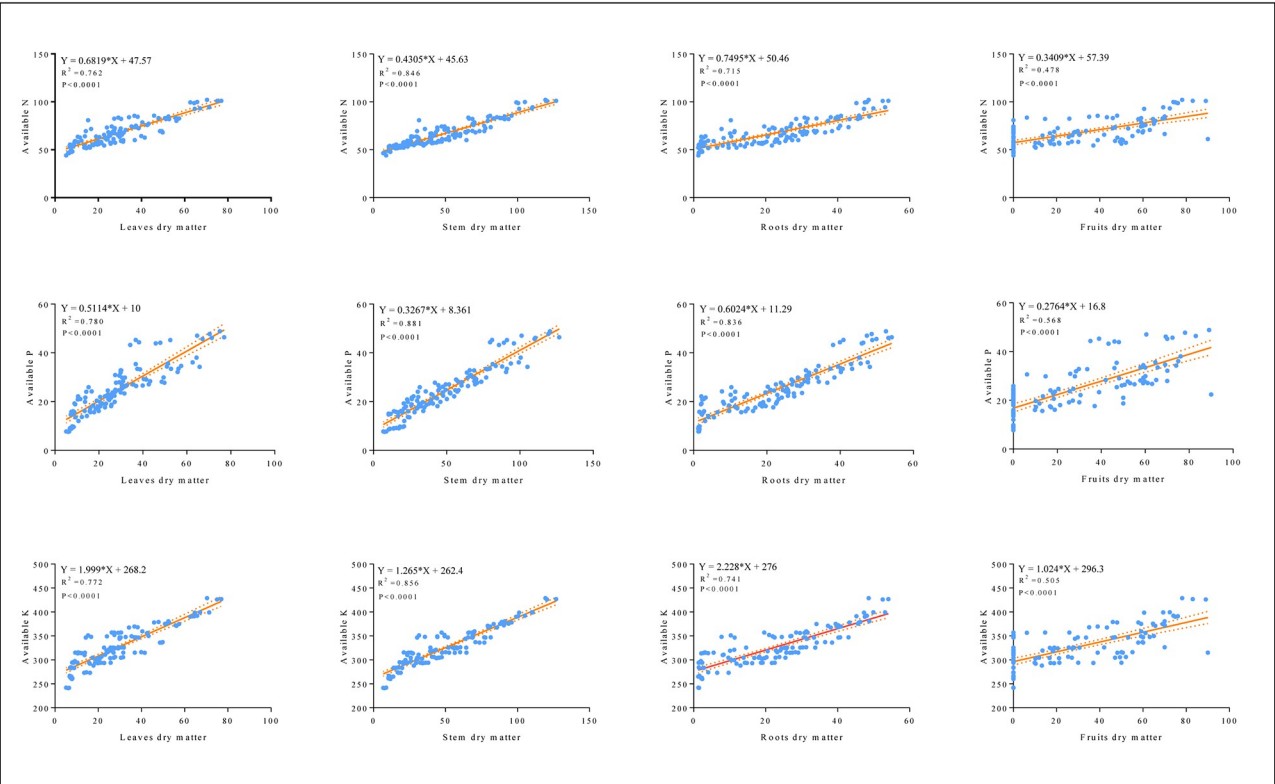

**Fig 3. The relationship between soil available nutrients and dry matter in different mixing ratio irrigation (2019).**

Fig 4b). Also, it was evident from the results that the highest root dry matter recorded in growth, boll and boll opening stage were 6.46g, 38.68g and 51.63g with T8 treatment, compared with the lowest values of 4.60g, 12.24g and 23.57g recorded in T7 treatment. A significant main and interactive effect were observed at boll and boll opening stages of roots dry matter ($p < 0.05$, Fig 4c). Fruits dry matter were produced in boll and boll opening stages and was substantially higher in T8 treatment compared with other applied treatments throughout the whole experiment. A significant main interaction effect on boll and boll opening stage while a significant interactive effect on boll stage were observed for fruits dry matter ($p < 0.05$, Fig 4d). Similarly, there was a strong positive correlation between soil available nutrients and dry matter accumulation in round irrigation scheme (Fig 5). Our result showed that the surface water irrigation along with different mixing ratio shows promising effect on increasing average dry matter accumulation in cotton plant.

### 3.3 Effects of different mixing ratio and round irrigation on soil available nutrients

Soil available nitrogen at a depth of 0–20 and 20–40 cm fluctuated greatly during the whole cotton growth period, and increased sharply after each irrigation time (Fig 6a and 6b). The concentration of N at 0–20 cm after 8th irrigation increased from 72.3 ± 0.15 mg/kg (U), to 101.3 ± 0.27 mg/kg (S), 77.1 ± 0.18 mg/kg (U:S = 1:1), 82.1 ± 0.21 mg/kg (U:S = 1:2) and 99.3 ± 0.37 mg/kg (U:S = 1:3), with an average increase rate of 40.1%, 6.6%, 13.5%, and 29.5%, respectively ($p < 0.05$, Fig 6a). The concentration of N at 20–40 cm after 8th irrigation in S treatment were increased by 37.4%, 7.1%, 20.0%, and 21.9% on average relative to U applied treatment ($p < 0.05$, Fig 6b). Trends of soil available nitrogen follows the order of S> U:

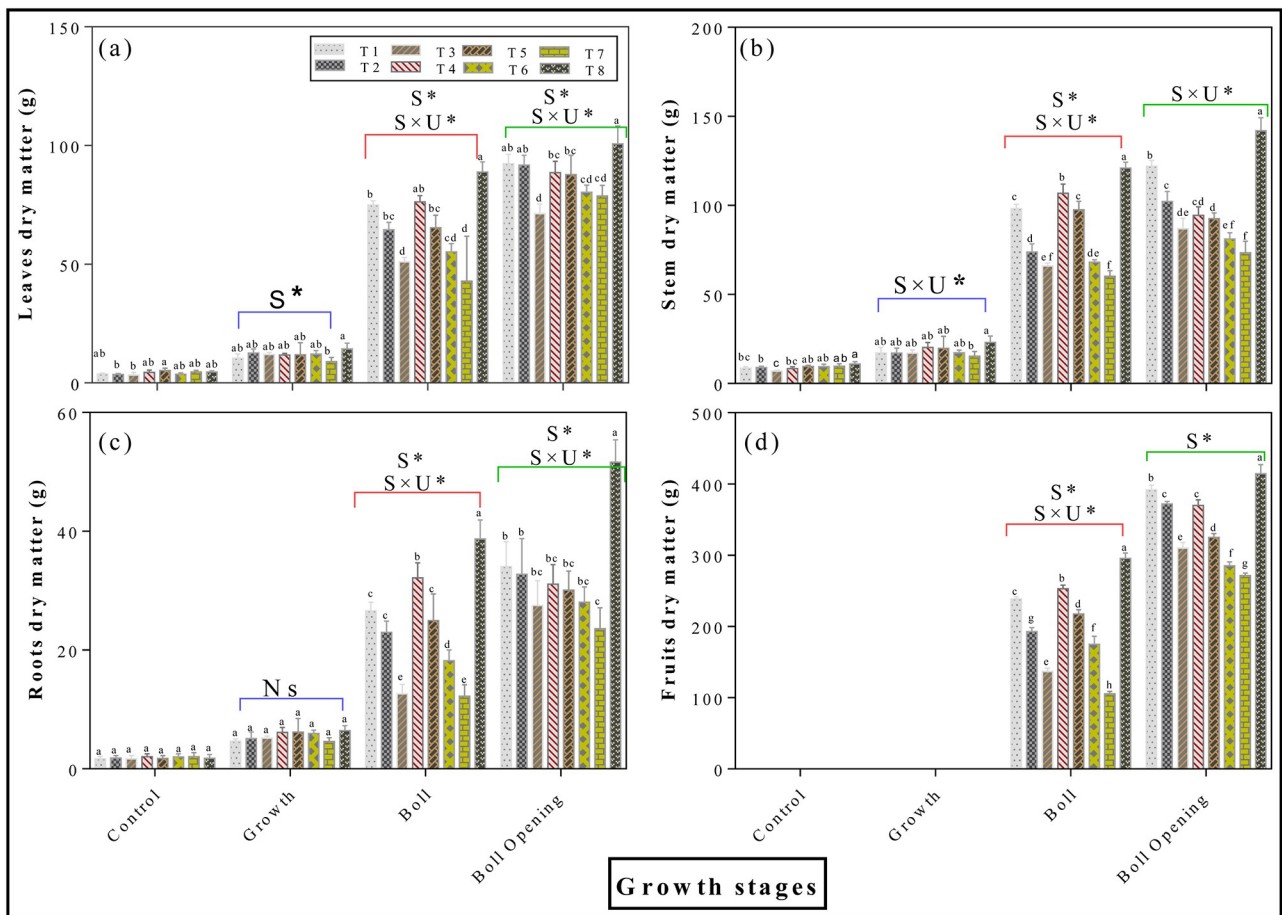

**Fig 4. Effects of round irrigation methods on dry matter accumulation: Fig (3a) represents leaves dry matter; Fig (3b) represents stem dry matter; Fig (3c) represent roots dry matter; and Fig (3d) represent fruits dry matter, respectively.** Data were presented as the mean ± standard error (SE) of three replicates at a significance level of p < 0.05. The inserted P values are from two-way ANOVAs.

S = 1:3> U:S = 1:2> U:S = 1:1>U. Moreover, the accumulation of P in different soil depths 0–20 and 20–40 cm was presented in (Fig 6c and 6d). The concentration of P at 0–20 cm after $8^{th}$ irrigation increased by 76.9%, and 33.8% %, respectively in S and U:S = 1:3 applied treatments compared with U treatment ($p < 0.05$, Fig 6c). Whilst at 20–40 cm, the amount of P in surface water application significantly increased from 12.6 ± 0.15 mg/kg (U), to 28.4 ± 0.27 mg/kg (S), 13.4 ± 0.18 mg/kg (U:S = 1:1), 13.3 ± 0.21 mg/kg (U:S = 1:2) and 22.6 ± 0.37 mg/kg U:S = 1:3 ($p < 0.05$, Fig 6d). The status of available potassium in soil at 0–20 and 20–40 cm remains parallel throughout the experiment (Fig 7a and 7b). Compared with all other applied treatments the surface water treatment showed maximum concentration of potash, however the difference between different applied treatments were negligible. Soil organic matter was significantly affected by different irrigation treatments (Fig 7c and 7d). However, at the start of the experiment the S treatment followed by U: S = 1:3 treatment shows potential promising effect at both 0–20 and 20–40 cm, whereas the difference between other treatments were negligible.

Furthermore, the concentration of soil available nutrients at a depth of 0-20cm and 20–40 cm in round irrigation method significantly affected by different applied treatments. The

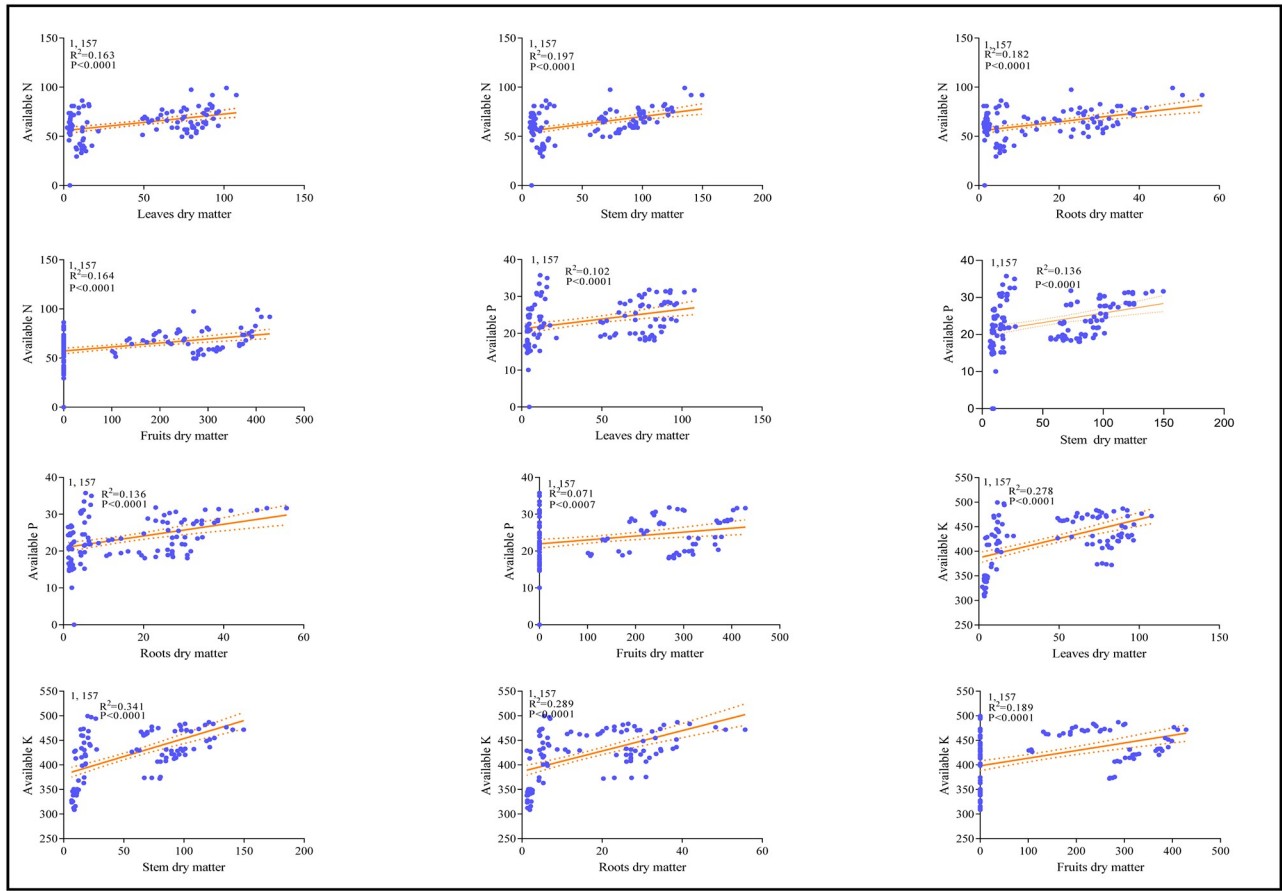

**Fig 5. The relationship between soil available nutrients and dry matter in round irrigation (2020).**

concentration of N recorded at 0–20 cm at different growth stages were 83.3±2.8 (growth stage), 79.01±1.84 (boll stage), 96.16±3.83 (boll opening stage) in T8, while in T7 the concentration of N was 36.1±5.9 (growth), 54.51±2.81 (boll), and 53.9±3.83 (boll opening) ($p < 0.05$, Fig 8a and 8b). The soil available P content significantly increased in the initial and final stages with T8 treatment throughout the experiment ($p < 0.05$, Fig 8c and 8d). Whereas, the status of available potassium in soil remains parallel throughout the experiment, the difference between different applied treatments were negligible ($p < 0.05$, Fig 9a and 9b). The maximum concentration of potash was determined in T8 treatment at growth, boll and boll opening stages. Soil organic matter was significantly affected by different irrigation treatments ($p < 0.05$, Fig 9c and 9d). However, at the start of the experiment surface water irrigation treatment shows promising effect at both 0–20 and 20–40 cm, while the difference between other treatments were negligible.

## 3.4 Effects of different irrigation methods on cotton yield

The effect of different mixing ratio irrigation on cotton yield is presented in Fig 10a. Cotton yield responded significantly to different water irrigation modes. For instance, the maximum cotton yield 6571 kg h$^{-1}$ was recorded in (S) treatment compared with the treatment of (U) (5492 kg h$^{-1}$), U: S = 1:1 treatment (5502 kg h$^{-1}$), U: S = 1:2 treatment (5873 kg h$^{-1}$) and U:

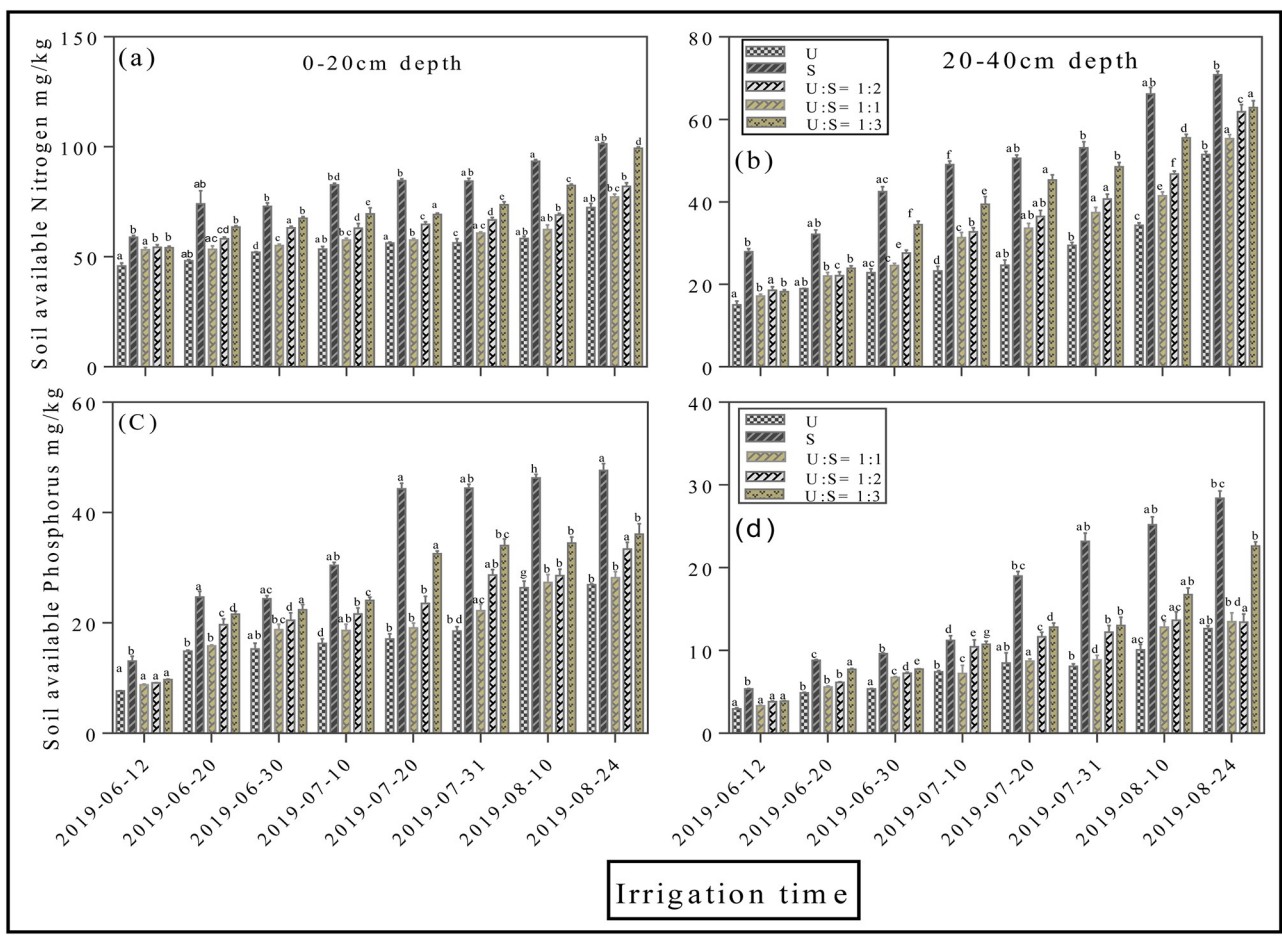

**Fig 6. Effects of different mixing ratio irrigation on soil available nitrogen (mg/kg) and soil available phosphorus (mg/kg): Fig (4a) and (4b) represents soil available nitrogen at depth of 0–20 cm and 20–40 cm; Fig (4c) and (4d) represents soil available phosphorus at depth of 0–20 cm and 20–40 cm.** Data were presented as the mean ± standard error (SE) of three replicates at a significance level of p < 0.05 (based on ANOVA).

S = 1:3 treatment (6111 kg h⁻¹). Likewise, round irrigation potentially effect cotton yield under different applied treatments (Fig 10b). For example, the increasing trend follow the order of T8 > T1 > T4 > T2 > T5 > T3 > T6> T7, which shows that surface water irrigation can effectively guarantee cotton production. The highest cotton yield was observed in T8 treatment in which surface water was supplied through all stages of cotton growth, followed by treatment T1 and T4 (Fig 10b). However, the lowest yield was recorded in treatment T7 in which underground water was supplied. Correlation and regression analysis was performed to explore the relationship between soil available nutrients and cotton yield (Fig 11). It can be seen from the figure that there is a negative correlation between soil available nutrients and cotton yield in different mixing ratio irrigation (2019), while a strong positive correlation was found in round irrigation (2020) between soil available nutrients and cotton yield.

## 4. Discussion

Water, the vital element for excessive irrigation, plays an extremely important role in agriculture. Currently agriculture is the largest water-consuming sector and approximately accounts for 60% of the total water resources consumption and has become a big concern for sustainable

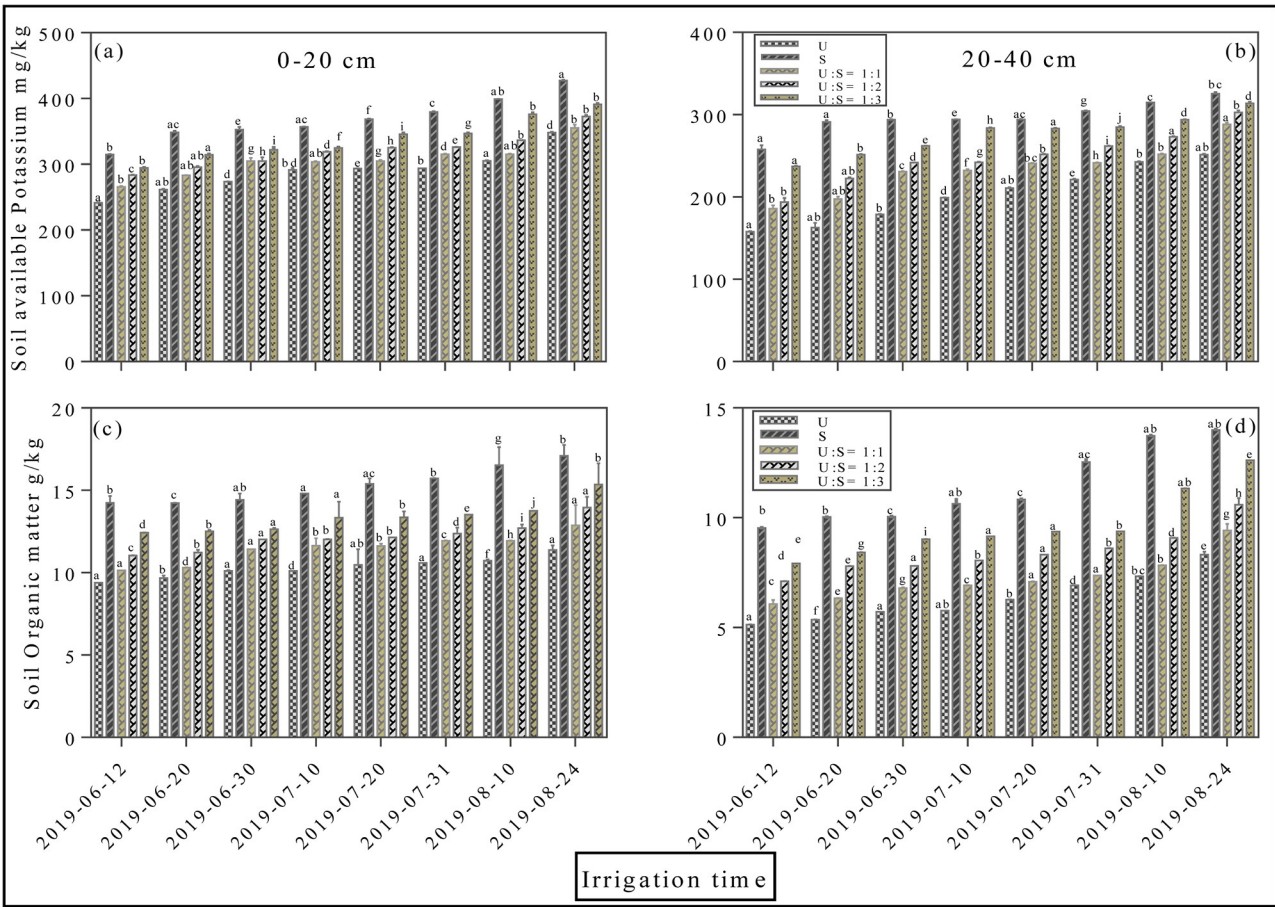

**Fig 7. Effects of different mixing ratio irrigation on soil available potassium (mg/kg) and soil organic matter (g/kg): Fig (5a) and (5b) represents soil available potassium at depth of 0–20 cm and 20–40 cm; Fig (5c) and (5d) represents soil organic matter at depth 0–20 cm and 20–40 cm.** Data were presented as the mean ± standard error (SE) of three replicates at a significance level of p < 0.05 (based on ANOVA).

crop production. Cotton (Gossypium hirsutum L.) production in Xinjiang strongly relay on sufficient irrigation conditions however water scarcity is one of the critical constraints for the sustainable production of cotton [20]. Meanwhile, surface water evaporation caused by high temperatures results in a severe water shortage leads to soil salinization, a lowered survival rate for crops, and slow development of local agriculture [25]. Similarly, increased usage of underground water for irrigation exacerbates the soil salinization problems, which significantly reduce crop yield [10, 11]. In this study, the application of surface water in both mixing ratio and round irrigation methods, significantly promotes dry matter accumulation (Figs 2–4), NPK uptake (Figs 6–9) and cotton yield (Figs 10 and 11), compared with all other applied treatments. Our obtained results are in line with the findings of previous published literature [26–28]. The superiority of surface water over underground water in calcareous soil may likely be due to the following reasons; in general surface water possess high temperature while the underground water temperature is quite low. Previous study has shown that well water irrigation with low temperature potentially inhibits the growth and development of jujube [29]. Likewise, several published literatures have shown that underground water irrigation (low temperature water) affect the growth, yield, dry matter accumulation and active developmental stages of grains plants such as peanuts, cucumber, and tomato [30–33]. For instance, a study

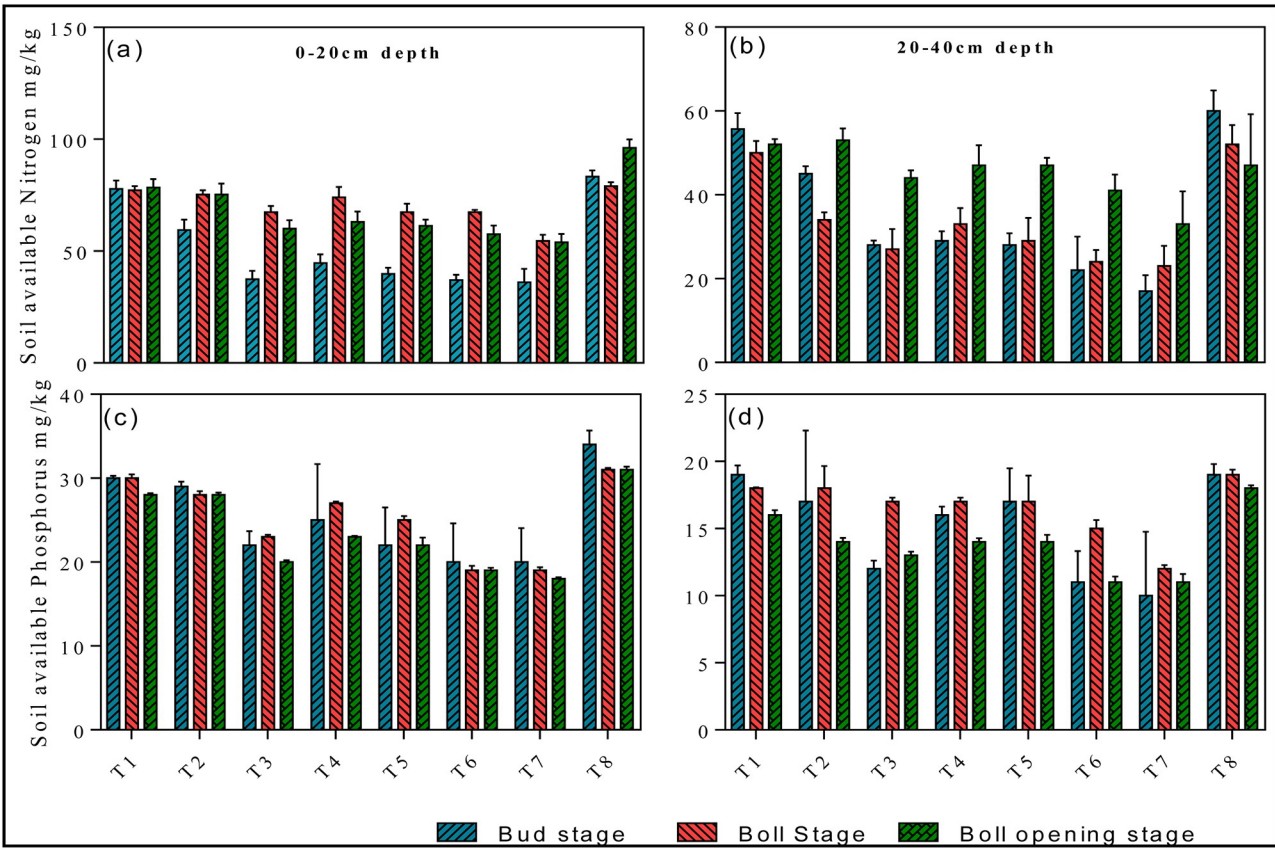

**Fig 8. Effects of round irrigation on soil available nitrogen (mg/kg) and soil available phosphorus (mg/kg): Fig (6a) and (6b) represents soil available nitrogen at depth of 0–20 cm and 20–40 cm; Fig (6c) and (6d) represents soil available phosphorus at depth of 0–20 cm and 20–40 cm.** Data were presented as the mean ± standard error (SE) of three replicates at a significance level of $p < 0.05$ (based on ANOVA).

conducted by Meng et al., (2016), noted that underground water irrigation significantly affects the growth and development of cotton plant. Similar results with the application of underground water irrigation is also obtained in this study. Furthermore, Deng et al., [34], also pointed out that underground water irrigation along with their low temperature properties significantly retarded the growth of vegetables and their photosynthetic developments. Consequently, the excessive usage of underground water irrigation results in the accumulation of toxic substances in soil which alternatively leads to reduction in plants and grains yields [35, 36]. The accumulation of salt can directly decrease soil nutrient efficiency by inhibiting microbial mineralization activity in soil [37]. Additionally, salinity can also indirectly affect soil nutrient cycling and efficiency by destroying soil physical structure [38–40]. On the flip side, the study of Zhang et al., 2002 [26] showed that alternate irrigation with surface fresh water can reduce soil salt content and increase cotton production which are in line with our findings. In this study surface water irrigation along with different mixing ratios irrigation also shows promising effects on cotton yield and dry matter accumulation when compared with underground water treatment alone. This is possibly due to when the underground water and surface water were mixed together, the temperature and salt content were changed. For example, a study conducted by Tao et al., 2014 [27], showed that mixed irrigation mode of brackish and fresh water with a salinity of 1.6g/L could achieve higher crop yield with better quality. Consistently, a study carried out by Wang et al., 2010 [28], showed that well and canal mixed

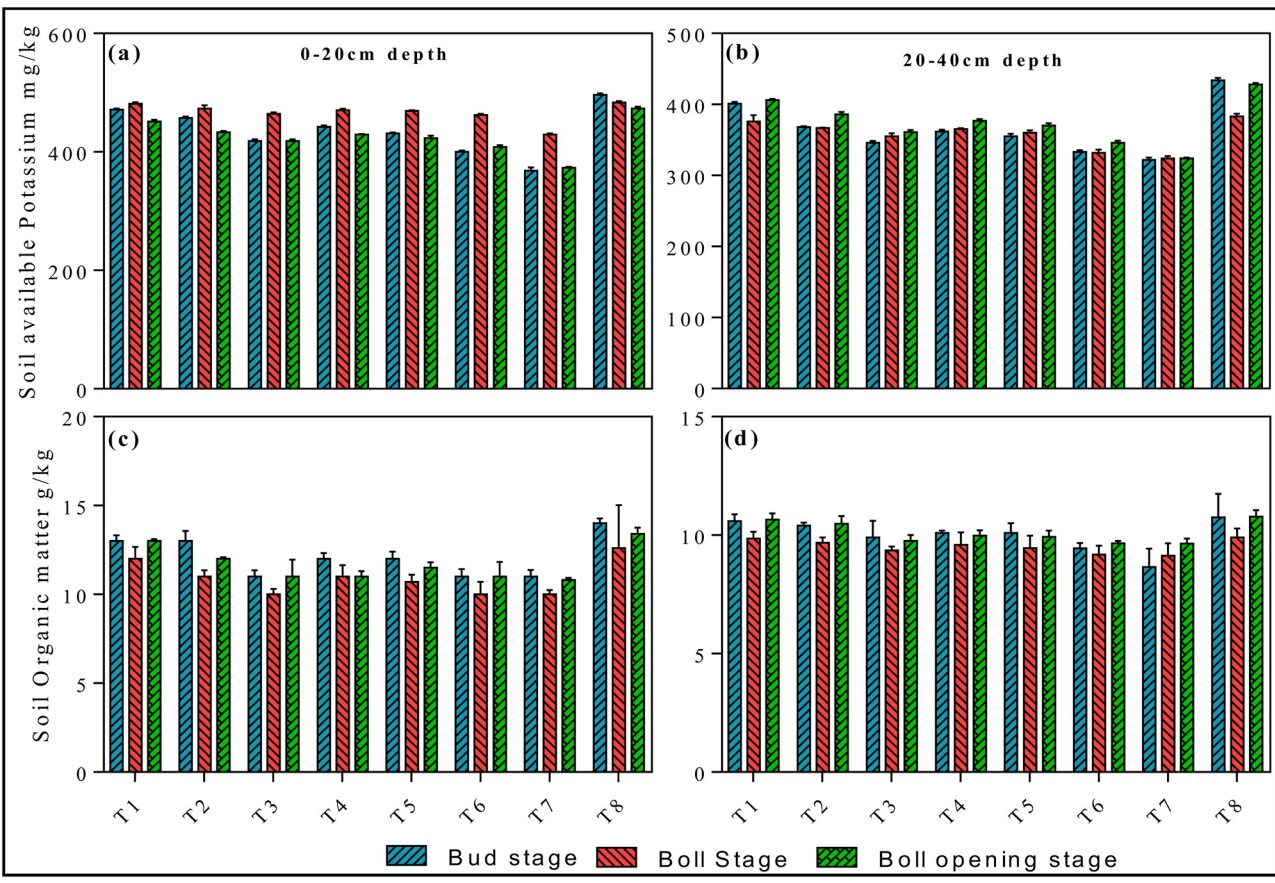

**Fig 9. Effects of round irrigation on soil available potassium (mg/kg) and soil organic matter (g/kg): Fig (7a) and (7b) represents soil available potassium at depth of 0–20 cm and 20–40 cm; Fig (7c) and (7d) represents soil organic matter at depth 0–20 cm and 20–40 cm.** Data were presented as the mean ± standard error (SE) of three replicates at a significance level of $p < 0.05$ (based on ANOVA).

irrigation could keep the salt balance of root soil even in relatively dry years. Whereas, well irrigation alone results in salt accumulation in roots of winter wheat and decreased the yield up to 20% to 30%. All these findings suggest that under-ground water irrigation possess negative effects on plant yield and growth whilst surface water irrigation and different mixing ratios irrigation significantly promote cotton yield NPK, uptake and dry matter accumulation.

## 5. Conclusions

It can be concluded that the application of surface water along with their different mixing ratios irrigation outcompete underground water irrigation in both different mixing ratio and round irrigations methods. A significant highest dry mater accumulation, NPK uptake and cotton yield at various stages i.e., growth stage, boll stage, and boll opening stage were always noted with surface water applied treatments compared with underground water treatment. The dry matter accumulation, NPK uptake, and cotton yield were suppressed more regularly by underground water treatments. Although the effects of surface water irrigation to agricultural soils on increasing cotton yield is promising. Nevertheless further study is needed by using edge cutting technologies to fully underpin the underlying mechanism of surface water irrigation and underground water irrigation and their interaction with soil particles in a wide-range of soil conditions.

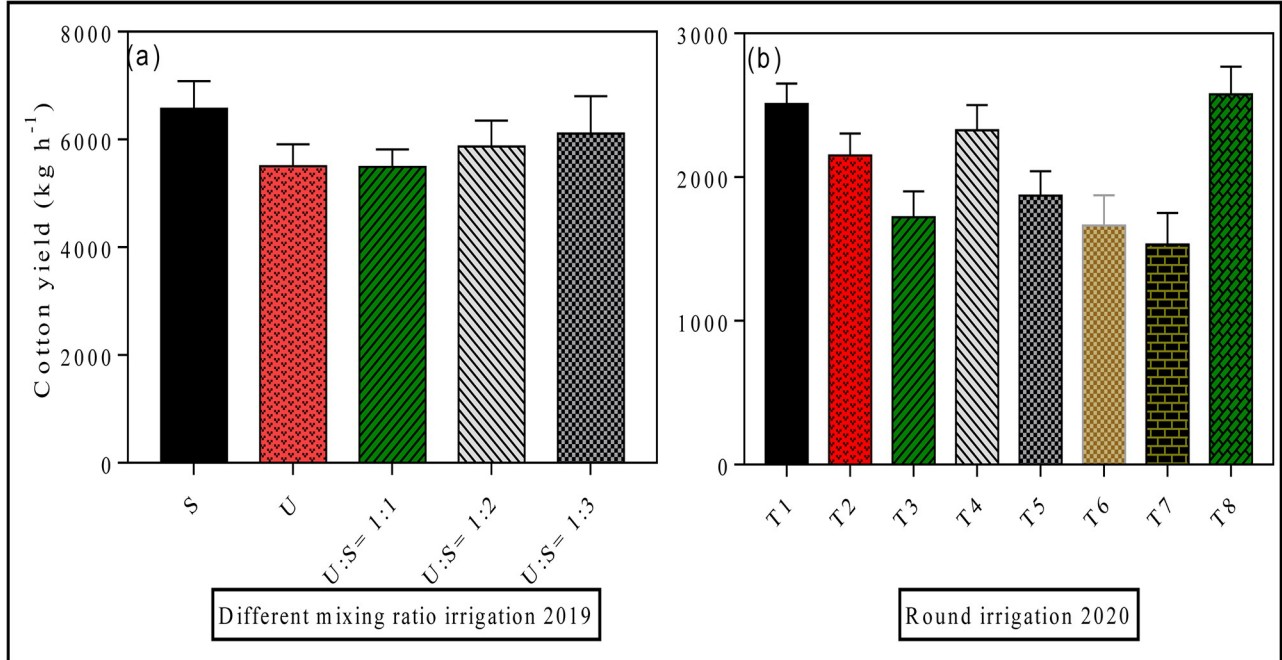

**Fig 10. Effects of different irrigation methods on cotton yield.** Data were presented as the mean ± standard error (SE) of three replicates at a significance level of p < 0.05 (based on ANOVA).

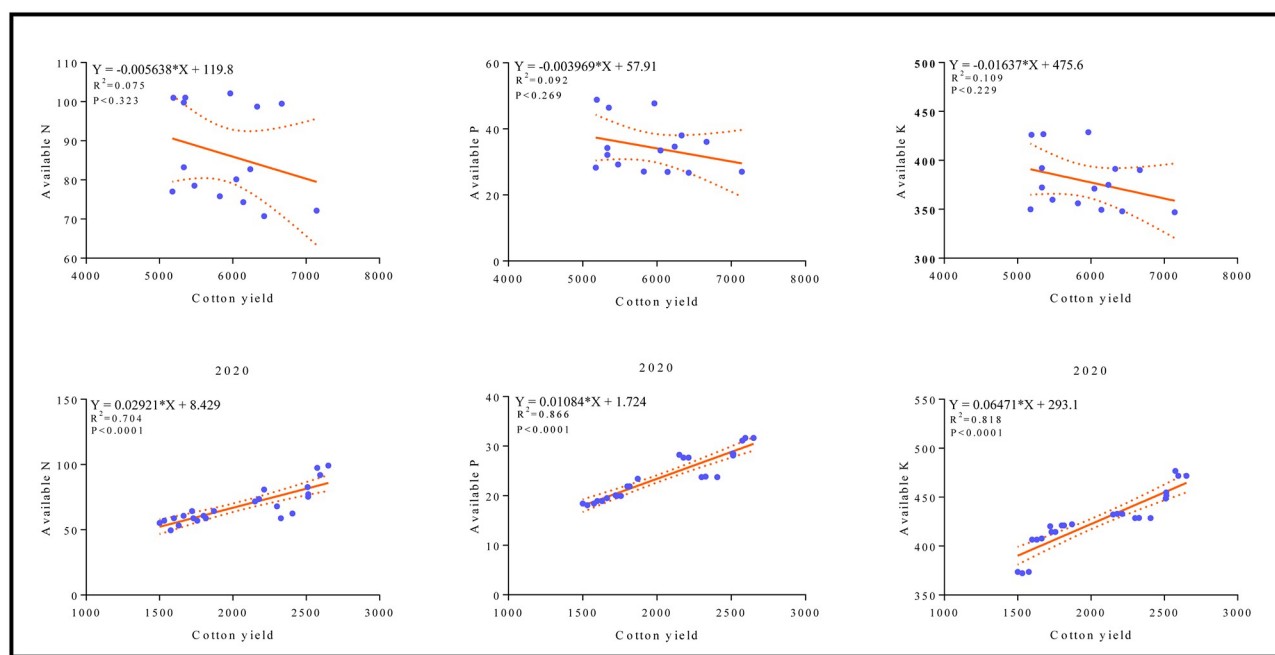

**Fig 11. The relationship between soil available nutrients and cotton yield.**

## Supporting information

**S1 Fig. Field pictures of the experimental scheme.**
(DOCX)

**S1 Data.**
(XLSX)

**S2 Data.**
(XLSX)

**S1 File.**
(PDF)

## Author Contributions

**Conceptualization:** Nihal Niaz, Cheng Tang.

**Data curation:** Nihal Niaz.

**Formal analysis:** Nihal Niaz.

**Funding acquisition:** Cheng Tang.

**Investigation:** Nihal Niaz.

**Methodology:** Nihal Niaz, Cheng Tang.

**Project administration:** Cheng Tang.

**Resources:** Cheng Tang.

**Software:** Nihal Niaz.

**Supervision:** Cheng Tang.

**Validation:** Cheng Tang.

**Visualization:** Cheng Tang.

**Writing – original draft:** Nihal Niaz.

**Writing – review & editing:** Nihal Niaz.

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
