## [Decision Letter · Decision Letter 0]

20 Apr 2022

PONE-D-22-03631Effect of surface water and underground water drip irrigation on Cotton growth & yield under two different irrigation schemes.PLOS ONE

Dear Dr. Tang,

Thank you for submitting your manuscript to PLOS ONE. After careful consideration, we feel that it has merit but does not fully meet PLOS ONE’s publication criteria as it currently stands. Therefore, we invite you to submit a revised version of the manuscript that addresses the points raised during the review process.

The manuscript  is kind of incomplete to deliver the novelty of the research work. The authors are asked to follow the comments of the review and answer them accordingly with appropriate documentation, data analysis, and interpretation of results in an improved manner.  

We look forward to receiving your revised manuscript.

Kind regards,

Rafiq Islam, Ph.D.

Academic Editor

PLOS ONE

Journal Requirements:

Reviewers' comments:

Reviewer's Responses to Questions

**Comments to the Author**

1. Is the manuscript technically sound, and do the data support the conclusions?

Reviewer #1: No

2. Has the statistical analysis been performed appropriately and rigorously? 

Reviewer #1: No

3. Have the authors made all data underlying the findings in their manuscript fully available?

Reviewer #1: No

4. Is the manuscript presented in an intelligible fashion and written in standard English?

Reviewer #1: Yes

5. Review Comments to the Author

Reviewer #1: I reviewed the manuscript entitled: Effect of surface water and underground water drip irrigation on Cotton growth & yield under two different irrigation schemes.

That seemed to be an interesting manuscript, but it wan not as I expected.

Corrections are specified in the attached file.

6. PLOS authors have the option to publish the peer review history of their article (what does this mean?). If published, this will include your full peer review and any attached files.

Reviewer #1: No

---

## [Author Response · Author response to Decision Letter 0]

20 May 2022

Dear Prof. Dr. Rafiq Islam 

Academic Editor

PLOS ONE,

Many thanks for your letter. We greatly appreciate the constructive suggestions and comments on our MS entitled “Effect of surface water and underground water drip irrigation on Cotton growth and yield under two different irrigation schemes" from both yourself and the reviewers. All those comments are valuable and very helpful for us to improve our manuscript. We extend our great appreciation for taking the time and efforts to provide such insightful guidance. We have taken a complete consideration to all reviewers’ comments as well as those suggestions from the editor’s and have made the corrections one by one in the revised version of our manuscript. The changes in the revised MS are marked in track change model.

 We sincerely hope the revised manuscript will be able to meet the requirement and will be finally accepted to publish on your journal of “PLOS ONE”. Of course, we are always available to provide ongoing changes to our manuscript if there is any further request either from you or from the reviews. 

Below please find the revised manuscript and the responses. Again, thank you and all the reviewers for your kinder assistance and we are looking forward to hearing from you at your earliest convenience.

Best wishes 

Many thanks again

On behalf of the all coauthors

Sincerely yours,

---

## [Editor Report · Decision Letter 1]

31 May 2022

PONE-D-22-03631R1Effect of surface water and underground water drip irrigation on cotton growth and yield under two different irrigation schemesPLOS ONE

Dear Dr. Tang,

Thank you for submitting your manuscript to PLOS ONE. After careful consideration, we feel that it has merit but does not fully meet PLOS ONE’s publication criteria as it currently stands. Therefore, we invite you to submit a revised version of the manuscript that addresses the points raised during the review process.

The authors were asked to address the reviewer's and academic editor's comments appropriately to improve the quality of the manuscript. Details are in the attached academic editor reviewed copy of the manuscript. 

We look forward to receiving your revised manuscript.

Kind regards,

Rafiq Islam, Ph.D.

Academic Editor

PLOS ONE

Additional Editor Comments (if provided):

The revised manuscript, in its current content, format, and interpretation, is unsuitable to accept for publication. The authors failed to address the concerns of the reviewer(s). I fact, none of the responses were realistic. It needs a total and thorough revision with appropriate statistical analysis, prestation of data in tables and figures, and interpretation of results.

The abstract should be focused and concise to summarize the whole work. Introduction needs to be written based on global context with connecting paragraph links to each other with a mission and vision.

The methods and materials section need in-depth explanation about the experimental design, factors (fixed vs. random), cultural practices and fertilization, analysis of soil and plant samples, appropriate statistical analysis, and presentation of data in significant digits with interactions. Treatments should be separated as factors such as Underground vs. Surface water, Growth stages, soil depth, and time. The ANOVA should be 2 x 2 (water sources and ratios), 2 x 2 x 3 (water sources and ratios, cotton growth, and time), and 2 x 2 x 3 x 2 (water sources and ratios, cotton growth, time x soil depth), respectively.

Randomized complete design block means there are site variability. The block effect should be shown and explained to justify the results.

Cleary explain the irrigation frequencies and timing, and volume, types and amount of fertilization, sampling of soil and plant samples, and methods of soil and plant analysis. The SOM cannot be determined by K2Cr2O7 oxidation method, it should be total organic carbon (TOC). When TOC multiplied with 1.724 then it will be SOM. Moreover, it should be “Perchloric acid.” Data need to present in mean + standard error of mean. Should add a section on water-use efficiency.

It would be good for the authors to consult a statistician to reorganize and analyse the data and interpret the results. Based on revised statistical data analysis, the results and discussion sections should be rewritten. The data presented in Table 5 to 8 were unrealistic, way too high for available NPK concentration during the season and at the end of the season. Reanalyze the samples or recalculate the data to verify the concentration. SOM contents do not really changed in a season!

Conclusions should be concise and focused based on results with a suggestion.
---

## [Author Response · Author response to Decision Letter 1]

14 Jul 2022

Academic Editor

PLOS ONE,

Many thanks for your letter. We greatly appreciate the constructive suggestions and comments on our MS entitled “Effect of surface water and underground water drip irrigation on Cotton growth & yield under two different irrigation schemes" from both yourself and the reviewers. All those comments are valuable and very helpful for us to improve our manuscript. We extend our great appreciation for taking the time and efforts to provide such insightful guidance. We have taken a complete consideration to all reviewers’ comments as well as those suggestions from the editor’s and have made the corrections one by one in the revised version of our manuscript. The changes in the revised MS are marked in track change model.

 We sincerely hope the revised manuscript will be able to meet the requirement and will be finally accepted to publish on your journal of “PLOS ONE”. Of course, we are always available to provide ongoing changes to our manuscript if there is any further request either from you or from the reviews. 

Below please find the revised manuscript and the responses. Again, thank you and all the reviewers for your kinder assistance and we are looking forward to hearing from you at your earliest convenience.

Best wishes 

Many thanks again

On behalf of the all coauthors

Sincerely yours,

---

## [Editor Report · Decision Letter 2]

1 Aug 2022

PONE-D-22-03631R2Effect of surface water and underground water drip irrigation on cotton growth and yield under two different irrigation schemesPLOS ONE

Dear Dr. Tang,

Thank you for submitting your manuscript to PLOS ONE. After careful consideration, we feel that it has merit but does not fully meet PLOS ONE’s publication criteria as it currently stands. Therefore, we invite you to submit a revised version of the manuscript that addresses the points raised during the review process.

 While the revised manuscript quality has improved; it still needs a thorough editing for correcting grammatical mistakes and spelling to improve the quality of the research findings. As abstract is the summary of the whole research, it needs an extensive review and editing. The authors should simplify the sentences for better flow of understanding. Moreover, the statistical analysis section is unclear to understand the variables, analysis and interpretation. Data presented in the Tables especially Table 4 is very confused, such as m^3^/667 h^-1^. It could be just m^3^/ha, and in the Materials and methods section, this could be explained as total volume of water applied for irrigating the cotton was … m^3^/ha. This needs to be very clearly explained in the methodology.

We look forward to receiving your revised manuscript.

Kind regards,

Rafiq Islam, Ph.D.

Academic Editor

PLOS ONE
---

## [Author Response · Author response to Decision Letter 2]

15 Aug 2022

Dear Professor Dr. Rafiq Islam

Academic Editor

PLOS ONE,

Many thanks for your letter. We greatly appreciate the constructive suggestions and comments on our MS entitled “Effect of surface water and underground water drip irrigation on Cotton growth & yield under two different irrigation schemes" from both yourself and the reviewers. All those comments are valuable and very helpful for us to improve our manuscript. We extend our great appreciation for taking the time and efforts to provide such insightful guidance. We have taken a complete consideration to all reviewers’ comments as well as those suggestions from the editor’s and have made the corrections one by one in the revised version of our manuscript. The changes in the revised MS are marked in track change model.

 We sincerely hope the revised manuscript will be able to meet the requirement and will be finally accepted to publish on your journal of “PLOS ONE”. Of course, we are always available to provide ongoing changes to our manuscript if there is any further request either from you or from the reviews. 

Below please find the revised manuscript and the responses. Again, thank you and all the reviewers for your kinder assistance and we are looking forward to hearing from you at your earliest convenience.

Best wishes 

Many thanks again

On behalf of the all coauthors

Sincerely yours,

---

## [Editor Report · Decision Letter 3]

31 Aug 2022

Effect of surface water and underground water drip irrigation on cotton growth and yield under two different irrigation schemes

PONE-D-22-03631R3

Dear Dr. Tang,

We’re pleased to inform you that your manuscript has been judged scientifically suitable for publication and will be formally accepted for publication once it meets all outstanding technical requirements.

Kind regards,

Rafiq Islam, Ph.D.

Academic Editor

PLOS ONE
---

## [Editor Report · Acceptance letter]

19 Sep 2022

PONE-D-22-03631R3 

Effect of surface water and underground water drip irrigation on cotton growth and yield under two different irrigation schemes 

Dear Dr. Tang:

I'm pleased to inform you that your manuscript has been deemed suitable for publication in PLOS ONE. Congratulations! Your manuscript is now with our production department. 

Kind regards, 

on behalf of

Dr. Khandakar R. Islam 

Academic Editor

PLOS ONE